# Epidemiology of Respiratory Syncytial Virus-Related Hospitalization Over a 5-Year Period in Italy: Evaluation of Seasonality and Age Distribution Before Vaccine Introduction

**DOI:** 10.3390/vaccines8010015

**Published:** 2020-01-04

**Authors:** Federica Barbati, Maria Moriondo, Laura Pisano, Elisa Calistri, Lorenzo Lodi, Silvia Ricci, Mattia Giovannini, Clementina Canessa, Giuseppe Indolfi, Chiara Azzari

**Affiliations:** Department of Health Sciences, University of Florence and Meyer Children’s Hospital, Viale Pieraccini 24, 50139 Florence, Italy; federica.barbati@unifi.it (F.B.); maria.moriondo@meyer.it (M.M.); laura.pisano@unifi.it (L.P.); elisa.calistri@unifi.it (E.C.); silvia.ricci@meyer.it (S.R.); mattia.giovannini@unifi.it (M.G.); clementina.canessa@meyer.it (C.C.); giuseppe.indolfi@unifi.it (G.I.); chiara.azzari@unifi.it (C.A.)

**Keywords:** respiratory syncytial virus, hospitalization, epidemiology, seasonal trend, intensive care, vaccine, children

## Abstract

Respiratory Syncytial Virus (RSV) is associated with most of the acute viral respiratory tract infections causing hospitalization with a peak during the first months of life. Many clinical trials of RSV vaccine candidates are being carried out. The aim of this study was to obtain epidemiologic information to give suggestions on target populations and prevention strategies before the introduction of new vaccines or monoclonal antibodies. We retrospectively evaluated, over a 5-year period (September 2014–August 2019), a population of hospitalized Italian children aged 0–6 years with a laboratory confirmed diagnosis of RSV infection. Risk factors, seasonality of RSV infection, distribution according to age, cases of coinfections and reinfections and cases needing Intensive Care Unit were evaluated. Hospitalizations due to RSV were 624 in the period under study. The peak was found between November and April, with 80.4% of cases recorded between December and February. 62.5% of cases were found in children under three months of age and 41% in children under 30 days old. The need for intensive care was associated with younger ages, with 70.9% of cases in children below three months of age. Unless the incoming vaccines demonstrate a strong herd protection effect, preventive strategies should be aimed at newborns or at maternal immunization.

## 1. Introduction

Respiratory syncytial virus (RSV) is the most common respiratory agent in infants and young children worldwide [1]. It is the most frequent cause of hospitalization in children under two years of age, causing severe lower respiratory tract infections, such as pneumoniae and bronchiolitis, and it is associated with an increased risk of developing asthma and recurrent wheezing [2,3,4]. RSV is also the most common cause of postnatal infant mortality worldwide after malaria [5]. Furthermore, RSV infection is an important cause of morbidity in adults, particularly in immunocompromised individuals and in the elderly [6].

RSV infection is transmitted by direct inoculation or indirect contact with nasal or oral secretions. The clinical symptoms usually begin after an incubation period of 4 to 6 days with nasal congestion, rhinorrhea, coughing, a hoarse voice or fever. In some patients, the disease course may progress to the lower airways causing wheezing, tachypnea, jugular and intercostal retractions, hypoxemia or respiratory distress that needs hospital admission [7,8]. Pulmonary dysfunction related to RSV infection may last for many years; asthma, chronic wheezing and decreased respiratory functions have been found in children with a history of hospitalization due to RSV [9].

RSV is a seasonal virus and its epidemiology is extremely variable depending on the climatic setting, with the highest rates during the cold season in temperate climates and during the rainy season in tropical climates [10].

No specific treatment or vaccine prevention is available for RSV infections to date. Severe cases require supportive therapy, such as oxygen supplementation and hydration, and less frequently ventilatory support. The only antiviral agent licensed for the treatment of severe RSV infections is aerosolized Ribavarin, a synthetic guanosine analogue and broad-spectrum antiviral agent [11,12]. However, the use of Ribavarin has been limited to immunocompromised patients due to the cost of treatment, potential toxicity, such as teratogenic effects during pregnancy, and the need for hospital admission for prolonged aerosol administration [6].

Palivizumab (Synagis), a humanized monoclonal IgG1 antibody is at present the only market-approved protective strategy in infants who are high risk for severe RSV disease [13]. Palivizumab is administered monthly during the RSV season but its use is currently only approved for a subset of high-risk infants due to the high costs of the treatment.

The global burden of disease caused by RSV has become increasingly recognized, leading to many clinical trials of vaccine candidates. Several RSV immunoprophylaxis and vaccine candidates, including live attenuated, subunit, particle-based and live vectored agents are currently in development [14,15]. Identifying the part of the population with a higher risk of developing respiratory diseases that require hospitalization and identifying the seasonal patterns of RSV is important when planning strategies for prevention such as vaccination or monoclonal antibodies use. The aim of this study was to evaluate the period of the year and the age at which RSV was most frequent and severe in a pediatric hospitalized population in Tuscany, Italy, to obtain helpful epidemiologic information for the development of prevention strategies.

## 2. Materials and Methods

### 2.1. Study Design

This observational study retrospectively evaluated all children between 0 to 6 years of age included in the National Molecular Surveillance Register and who had been admitted to the hospitals of Arezzo, Empoli, Prato and Florence with a diagnosis of RSV infection from September 2014 to August 2019. The Molecular Surveillance Register was set up at the Immunology and Infectious Diseases Lab, Meyer Children’s Hospital, Florence, Italy, in 2006. Clinical and laboratory data were recorded using a standardized report form.

The study was approved by the Institutional Review Board. All data included in this study were obtained as part of routine clinical activity and evaluated retrospectively and anonymously in the study. For this reason, a specific approval by the ethical committee was not required. 

### 2.2. Laboratory Methods

The laboratory analysis was performed on a nasopharyngeal swab or on bronchoalveolar lavage fluid or on both. The test for RSV was requested alone or within an extended panel of Influenza-like-Illness (ILI) which among others also included Influenza virus and Rhinovirus. The presence of RSV in the samples were evaluated through Quantitative Reverse Transcription Polymerase Chain Reaction (RT-qPCR). Viral RNA was extracted from biological samples using QIAamp Viral RNA Mini Kit (Qiagen, Hilden, Germany) and reverse transcribed with High-Capacity RNA-to-cDNA Kit according to the manufacturer’s instructions. Positive and negative controls were used to validate each run. All tests were performed within routine clinical care using the primers and probes showed in Table 1 and amplification procedures previously described [16]. All reactions were performed in duplicate. If no increase in fluorescent signal was observed after 40 cycles, the sample was assumed to be negative.

## 3. Results

Data from 624 children with RSV infection were available (346 males, 55.4% and 278 females, 44.6%). All patients had respiratory distress requiring hospital admission and recovered after medical assistance. The laboratory analysis was performed on nasopharyngeal swabs in 602 cases, on bronchoalveolar lavage fluid in 11 cases and on both nasopharyngeal swabs and bronchoalveolar lavage fluid in 11 cases.

### 3.1. Morbidities

Prematurity (<37 weeks of gestational age) was detected in 153/624 patients (24.5%) and 6/624 (0.96%) had a diagnosis of bronchodysplasia. Of these 153 preterm infants, 111/153 (72.5%) were late preterm (34–37 weeks), 20/153 (13.1%) were moderate preterm (32–33 + 6 weeks), 13/153 (8.5%) were very preterm (28–31 + 6 weeks) and 9/153 (5.9%) were extremely preterm (<28 weeks). 

Additional underlying risk factors for RSV infection included 31 cases (4.97%) with congenital heart diseases, 28 (4.49%) with neurological impairment, 5 (0.8%) with immunological disease, 6 (0.96%) with oncological disorders and 4 (0.64%) with cystic fibrosis.

### 3.2. Seasonal Trend of Respiratory Syncytial Virus Infection in Tuscany

The data on each individual season allowed us to define the trend of RSV infection in the Tuscan paediatric population between September 2014 and August 2019. The epidemic period of RSV starts in late autumn (November), has a peak in winter (January) and has a variable end in early spring (April). 502/624 cases (80.4%) were recorded between December and February and 584/624 cases (93.6%) were recorded between December and March. The details of the seasonal trend of RSV infection is shown in Figure 1. 

The study also showed a progressive increase in the number of positive cases recorded during the five seasons examined in parallel with the progressive increase in the number of investigations requested by clinicians for etiological analysis of ILI. The details of the laboratory investigations requested and the RSV positive cases are shown in Table 2.

### 3.3. Distribution of Registered Cases Based on Age

The distribution of recorded hospitalized cases according to age demonstrated that 509/624 patients (81.6%) were under 1-year-old, 390/624 (62.5%) were under 3 months and 256/624 (41%) were under 30 days old.

Of the 509 patients under 1 year, 390 (76.6%) were between 0 and 3 months old and 256 (50.3%) were under 1 month old. The details of the distribution of the cases according to age is shown in Figure 2. 

### 3.4. Cases Admitted to the Intensive Care Unit

Of the 624 hospitalized children included in the study, 103 (16.5%) required admission to the Intensive Care Unit (ICU); 89/103 (86.4%) were under 1-year-old, 73/103 (70.9%) were under 3 months and 47/103 (45.6%) were under 1 month. The cumulative number of cases admitted to the Intensive Care Unit is shown in Figure 3.

### 3.5. Reinfections and Coinfections

Reinfection with subsequent rehospitalization was observed in five infants (0.8%). In one case the reinfection was more severe requiring admission to the ICU. Two of these five children had severe comorbidities: leukaemia in one case and primary immunodeficiency in the other.

In 49% of RSV positive cases the only laboratory investigation requested was the RSV PCR, while for the remaining 51% (*n* = 320) one or all of the ILI panel’s illnesses were also requested. In those cases, 2.2% of coinfection was found (two cases of Rhinovirus infection, one of Influenza B and four of Influenza A H3N2). Only in one case (one of the coinfection with Rhinovirus) the clinical presentation was more severe requiring admission to the ICU.

## 4. Discussion

The study showed that morbidities play an important role regarding the risk of RSV infection. In more than 30% of the patients at least one underlying condition was detected and in the majority of cases, prematurity. This rate was higher than the one reported in other studies [17,18].

Our data proved that the epidemic season in Italy starts in early November and lasts about six months, until April, with most of the cases (over 80%) recorded in the winter season between December and February. This seasonal pattern reflects the one identified by Janet et al. for the Temperate Northern Hemisphere [19]. RSV seasonality is highly dependent on geographic location. In temperate regions an annual seasonal pattern is predictably limited to 3–6 months during autumn, winter and early spring. Several explanations have been proposed, such as increased indoor crowding that enhances exposure to and the transmission of RSV, low temperatures present during winter that prolong the stability of RSV in fomites or low absolute humidity that increases the risk of RSV disease. In tropical regions, humidity and temperature play a different role, with studies suggesting that higher levels of humidity and stable temperatures allow large aerosol droplets to sustain RSV transmission all year around and in particular during the rainy season [20].

The last case recorded during the most recent season in Tuscany was found in July 2019. Our data did not show any case of RSV infection in this period of the year during the previous five epidemic seasons studied from 2014 to 2019. The child was a 10-month-old boy who had recently come from New Zealand, where the seasonal pattern of the RSV infection is different [19]. This confirms the presence of different epidemic seasons in opposite hemispheres and the necessity to carry out epidemiologic studies all over the world. Our data suggested that in Italy there was no reason to perform tests for RSV outside the epidemic season unless the patient came from a country where the seasonal pattern was different.

The number of positive cases increased progressively from the first to the last year of the study period. This can be related to the progressive increase in the number of tests requested by clinicians over the course of the study period, in addition to an improvement in laboratory techniques.

At present different prevention strategies are under study: vaccines aimed at pregnant women, therefore potentially offering passive immunity to newborns for a limited time period, vaccines aimed at infants three or more months old thereby administering active immune prophylaxis after that age and monoclonal antibodies, providing immune prophylaxis. In order to evaluate which option might be the most useful, epidemiological studies performed in different countries in the world are needed.

The study demonstrated that hospitalization due to RSV infection was typical of children under one year of age, with the highest frequency in children under three months of life. Intensive care support was strongly associated with younger ages, with over 80% of children under one year of age and over 70% of cases in children below three months of age accessing the ICU. Therefore, in order to reduce hospitalizations and severe cases requiring access to ICU, protection via early active immunization, like vaccination at birth, or a passive immunization with monoclonal antibodies or maternal immunization in the third trimester of pregnancy should be considered. An eventual vaccine administered later in life, for instance in the third month, would not be effective in reducing hospitalizations of children unless a strong herd-protection could be demonstrated. Therefore it is necessary to evaluate, with studies aimed at analysing the distribution of RSV in a non-hospitalized population and with adequate models, if RSV vaccinations performed later in life could reduce the hospitalization rate of very young children through herd immunity.

Viral coinfections can occur in children who are hospitalized with acute respiratory infections and bacterial infections may be commonly observed in the later stages of respiratory disease [21,22]. In our study the percentage of coinfection with Influenza or Rhinovirus was 2.2% of the cases in which the ILI panel was requested and only in 1/7 cases of coinfection, admission to the ICU was required; that proportion is not dissimilar to the proportion of ICU admission in children with no coinfection.

The RSV infection and the ILI have a similar clinical presentation and epidemic season, as we could see in a study recently conducted by Elhakim et al. [23]. In cases where RSV has had a negative result, it is imperative to search for other pathogens that are known to be associated with ILI. Given the similar clinical presentation, possible coinfection and the need to test for other pathogens if the RSV infection results are negative, requesting a full ILI-panel would be pertinent, especially if cheaper tests are to be developed.

The reinfections due to RSV were half of those reported in the study conducted in Austria by Resch et al. [18]. The rate of admission to ICU in our study was 16.5%, which was similar to what had been found in Austria and higher than the rate of 6.5% reported in an older study from Spain by Hervas et al. [24].

Vaccines have been licensed against the main causes of death associated with respiratory tract infections such as *Haemophilus influenzae* type B and *Streptococcus pneumoniae* and as a consequence RSV has remained one of the leading pulmonary causes of death for which no vaccine is yet available [25]. Lots of pharmaceutical companies are presently working on the development of new monoclonal antibodies and vaccines so that RSV infection could be prevented in the risk categories. Different vaccine strategies may be appropriate for heterogeneous target populations such as at-risk infants, school-age children, adult caregivers or the elderly [15]. It is important to carry out studies in different countries and involving the whole population to decide the right target for prevention strategies.

## 5. Conclusions

RSV is one of the major remaining common challenges in infectious diseases and a leading cause of hospitalization among young children, in particular in preterm infants. Our study is the largest RSV epidemiological study in Italy and covers a time period of 5 years; it demonstrates that RSV infection is more frequent in the period of the year between December and February and in children under three months of age. These data are necessary to plan preventive strategies, both in the case of passive immunization with recombinant monoclonal antibodies and in the case of either maternal or infant active immunization.

## Figures and Tables

**Figure 1 vaccines-08-00015-f001:**
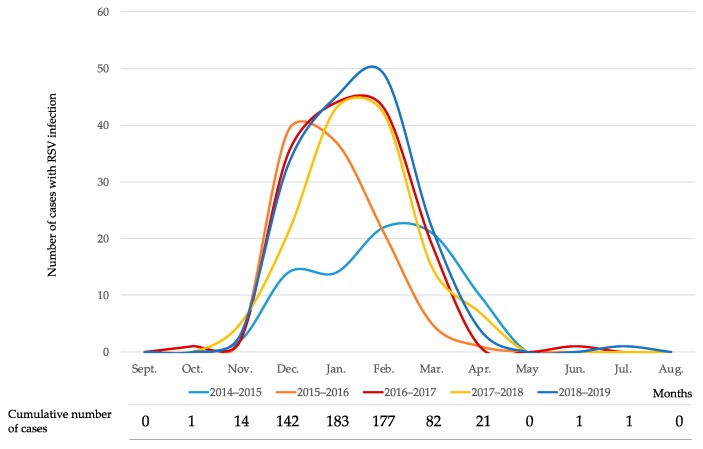
Seasonal trend of respiratory syncytial virus (RSV) infection in Tuscany in each single study year and cumulative number of cases in the 5-year study period.

**Figure 2 vaccines-08-00015-f002:**
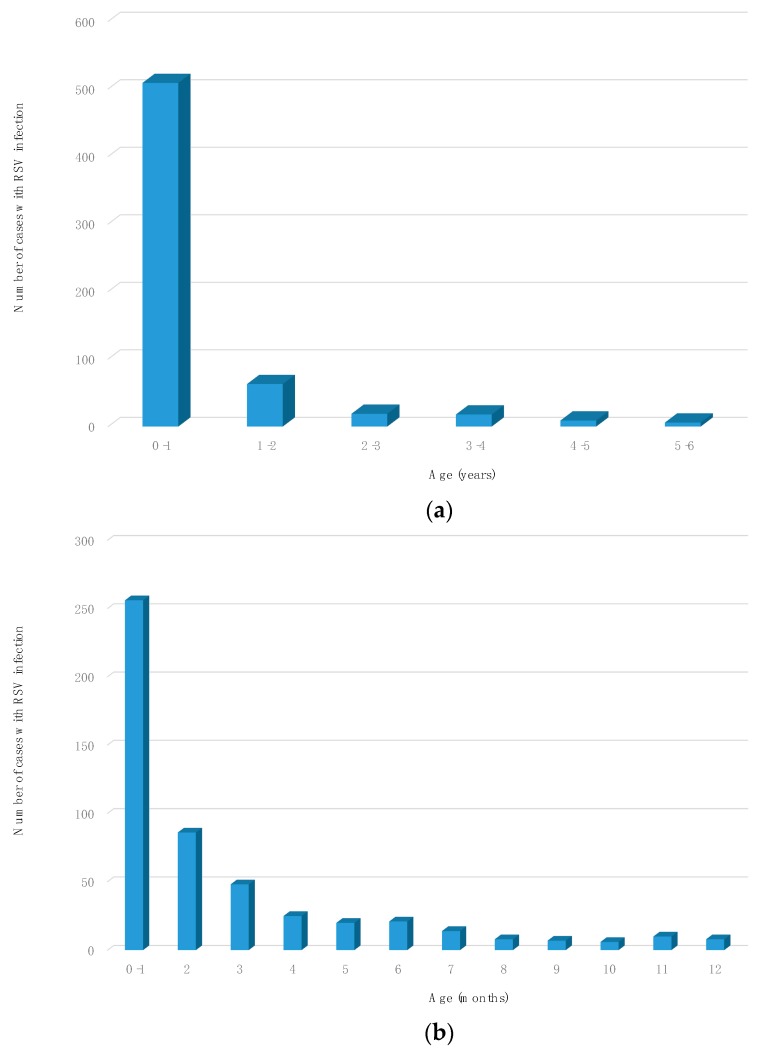
Number of cases of RSV infection: (**a**) cases divided by age expressed in years; (**b**) cases between 0–1 year-old divided by age expressed in months.

**Figure 3 vaccines-08-00015-f003:**
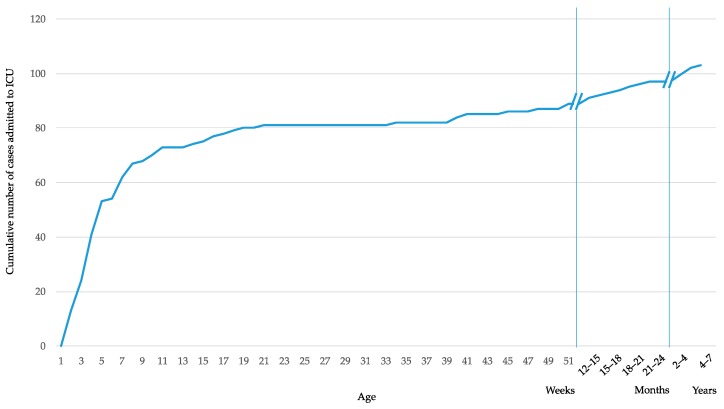
Cumulative number of cases admitted to the Intensive Care Unit.

**Table 1 vaccines-08-00015-t001:** Primer and probe sequence for Quantitative Reverse Transcription Polymerase Chain Reaction (RT-PCR).

Gene	RSV Matrix Protein Gene
Forward primer (5′-3′)	GGAAACATACGTGAACAARCTTCA
Reverse primer (5′-3′)	CATATTGTWAGTGATGCAGGATCAT
Probe (5′-3′)	FAM-AAGGCTCCACATACACAGCTGCTGT-TAMRA

**Table 2 vaccines-08-00015-t002:** Number of investigations requested, positive cases and % of positive cases divided by epidemic season.

Epidemic Season	Investigations Requested	RSV Positive Cases	% of Positive Cases
**2014–2015**	235	83	35.3
**2015–2016**	256	105	41
**2016–2017**	356	146	41
**2017–2018**	385	133	34.5
**2018–2019**	395	157	39.7
**Total**	1627	624	38.4

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
