# Peer review of "Epidemiology of Respiratory Syncytial Virus-Related Hospitalization Over a 5-Year Period in Italy: Evaluation of Seasonality and Age Distribution Before Vaccine Introduction"

_vaccines, 2020, doi:10.3390/vaccines8010015_

Round 1

Reviewer 1 Report

In their paper: “Epidemiology of Respiratory Syncytial Virus-related  Hospitalization over a 5-year period in Italy:  Evaluation of seasonality and age distribution before vaccine introduction” the authors perform a retrospective study in 3 hospitals in Tuscany to describe the seasonal distribution of, and need for, ICU care of children admitted to the hospital for RSV infection.  The study is limited in scope, and confirms the existing epidemiological data. References of previous studies should be included.

Although the authors describe the collection of clinical and laboratory data, in their paper clinical presentation, predisposing factors and final outcomes are missing.

The methods of detection are excessively detailed, yet it is unclear if these are approved diagnostics performed for routine clinical care or for research.  

There is no comment on the role of coinfection on the outcome.

The authors also talk about RSV vaccines as if this is available however, this vaccine is not yet approved. Clinical trials are undergoing and results are not available.

While epidemiological studies are important, it is also unlikely that the the results of this study by itself will affect recommendation on vaccine use. These statements should be revised

Author Response

Dear Reviewer, we are thankful for all the comments and suggestions. We have tried to clarify all the points and modified the manuscript accordingly. 

In the response we referred to the lines' numbers of the manuscript version with "track changes".

Reviewer 2 Report

The authors provided the original 5-year retrospective analysis of RSV infection in children of 0-6 yrs. old. The manuscript has unique content and clearly written. However, some minor modifications are required.

TITLE

(1). Capitalization in the MS title should be consistent throughout the title.

ABSTRACT

(2). The connection between epidemiologic analysis of RSV infection in children and RSV vaccines that are not available on the market should be clarified.

KEYWORDS

(3). How justified is adding a keyword ‘vaccine’? The authors did not demonstrate a direct connection between their results and the urgent need to protect children with vaccines.

MATERIALS AND METHODS

(4). Lane 68. ‘national’ should be capitalized.

RESULTS

(5). Fig.1. I think that Fig.1b is more or less repetition of Fig.1a. I would recommend removing Fig.1b and place a one-row table under Fig.1a which will contain a cumulative number of RSV infection cases exactly under appropriate months in Fig.1a horizontal axis legends. I would do so but this is up to the authors.

(6). Fig.1a,b. Please, add axis titles.

(7). If Fig.1b remains in the MS, please replace ‘Dic’ with ‘Dec’ (see horizontal axis legends).

(8). Fig.2a,b. I would recommend removing colored squares that designate different ages and add these specific ages directly as horizontal axis legends as it was done in Fig.3.

(9). Fig.3, horizontal axis legends. Please, remove ‘of life’ from ‘Years of life’, but add axis title ‘age.’

(10). Lane 116, Table 2, etc. I have found in a number of publications regarding RSV infection the term ‘RSV epidemic season.’ Please, clarify - does the RSV cause epidemics or local outbreaks? We do know that seasonal flu viruses cause epidemics. What the difference between flu epidemic season and RSV epidemic season? Is this just generally accepted terminology and in real life RSV cause only outbreaks during the so-called RSV epidemic season?

DISCUSSION

(11). The development and necessity of using RSV vaccines are good points for the discussion. However, in the current form, the discussion of vaccines seems like ‘a foreign body.’ The authors discussed in detail the need to protect children with RSV vaccines; they described different types of potential RSV vaccines etc. Clarification required why it is necessary to include this part in the Discussion if this issue is not addressed in the Results, the commercial vaccines are not available yet and the main aim of the study was obtaining epidemiologic information on RSV infection in children.

CONCLUSION

(12). The authors concluded that RSV infection is more frequent in the period of December-February and in children under three months of age. I am not sure that this is a unique finding and afraid that its novelty is limited. I think the conclusion should be rewritten to clarify the uniqueness of their findings. Maybe, it should be concretized that this is the first study in Italy (if so) or something else.

Author Response

Dear Reviewer, we are thankful for all your suggestions. We have provided a point-by-point response and modified the manuscript accordingly.

In the response we referred to the lines' numbers of the manuscript version with "track changes".
